# A Study on the Anti-NF-κB, Anti-*Candida*, and Antioxidant Activities of Two Natural Plant Hormones: Gibberellin A4 and A7

**DOI:** 10.3390/pharmaceutics14071347

**Published:** 2022-06-25

**Authors:** Bruno Dias Nani, Pedro Luiz Rosalen, Josy Goldoni Lazarini, Janaína de Cássia Orlandi Sardi, Diego Romário-Silva, Leonardo Pereira de Araújo, Mateus Silva Beker dos Reis, Isadora Breseghello, Thiago Mattar Cunha, Severino Matias de Alencar, Nelson José Freitas da Silveira, Marcelo Franchin

**Affiliations:** 1Department of Physiological Sciences, Piracicaba Dental School, University of Campinas, Piracicaba 13414-903, SP, Brazil; bd.nani@yahoo.com.br (B.D.N.); pedro.rosalen@unifal-mg.edu.br (P.L.R.); josy662@hotmail.com (J.G.L.); 2Graduate Program in Biological Sciences, Federal University of Alfenas (Unifal-MG), Alfenas 37130-001, MG, Brazil; 3Program on Integrated Dental Sciences, Faculty of Dentistry, University of Cuiabá, Cuiabá 78065-900, MT, Brazil; janasardi@gmail.com (J.d.C.O.S.); diegoromarioo@gmail.com (D.R.-S.); 4School of Pharmaceutical Sciences, Federal University of Alfenas (Unifal-MG), Alfenas 37130-001, MG, Brazil; leonardo.araujo@sou.unifal-mg.edu.br; 5School of Dentistry, Federal University of Alfenas (Unifal-MG), Alfenas 37130-001, MG, Brazil; mateus.beker@sou.unifal-mg.edu.br (M.S.B.d.R.); isadorabreseghello@hotmail.com (I.B.); 6Center for Research in Inflammatory Diseases (CRID), University of São Paulo, Ribeirão Preto 14049-900, SP, Brazil; thicunha@usp.br; 7Department of Agri-Food Industry, Food, and Nutrition, “Luiz de Queiroz” College of Agriculture, University of São Paulo, Piracicaba 13418-900, SP, Brazil; 8Laboratory of Molecular Modeling and Computer Simulation-MolMod-CS, Institute of Chemistry, Federal University of Alfenas, Alfenas 37130-001, MG, Brazil; nelsonjfs@gmail.com

**Keywords:** terpenoids, plant hormones, *Candida albicans*, macrophages, organic natural products

## Abstract

Introduction: Gibberellins (GA) are terpenoids that serve as important plant hormones by acting as growth and response modulators against injuries and parasitism. In this study, we investigated the in vitro anti-NF-κB, anti-*Candida*, and antioxidant activity of gibberellin A4 (GA4) and A7 (GA7) compounds, and further determined their toxicity in vivo. Methods: GA4 and GA7 in vitro toxicity was determined by MTT method, and nontoxic concentrations were then tested to evaluate the GA4 and GA7 anti-NF-κB activity in LPS-activated RAW-luc macrophage cell culture (luminescence assay). GA4 in silico anti-NF-κB activity was evaluated by molecular docking with the software “AutoDock Vina”, “MGLTools”, “Pymol”, and “LigPlot+”, based on data obtained from “The Uniprot database”, “Protein Data Bank”, and “PubChem database”. The GA4 and GA7 in vitro anti-*Candida* effects against *Candida albicans* (MYA 2876) were determined (MIC and MFC). GA7 was also evaluated regarding the viability of *C. albicans* preformed biofilm (microplate assay). In vitro antioxidant activity of GA4 and GA7 was evaluated against peroxyl radicals, superoxide anions, hypochlorous acid, and reactive nitrogen species. GA4 and GA7 in vivo toxicity was determined on the invertebrate *Galleria mellonella* larvae model. Results: Our data show that GA4 at 30 µM is nontoxic and capable of reducing 32% of the NF-κB activation on RAW-luc macrophages in vitro. In vitro results were confirmed via molecular docking assay (in silico), since GA4 presented binding affinity to NF-κB p65 and p50 subunits. GA7 did not present anti-NF-κB effects, but exhibited anti-*Candida* activity with low MIC (94 mM) and MFC (188 mM) values. GA7 also presented antibiofilm properties at 940 mM concentration. GA4 did not present anti-*Candida* effects. Moreover, GA4 and GA7 showed antioxidant activity against peroxyl radicals, but did not show scavenging activity against the other tested radicals. Both compounds did not affect the survival of *G. mellonella* larvae, even at extremely high doses (10 g/Kg). Conclusion: Our study provides preclinical evidence indicating that GA4 and GA7 have a favorable low toxicity profile. The study also points to GA4 and GA7 interference with the NF-κB via, anti-*Candida* activity, and a peroxyl radical scavenger, which we argue are relevant biological effects.

## 1. Introduction

Gibberellins (GA) are a large group of tetracyclic diterpenoid carboxylic acids that serve as important plant hormones by acting as growth modulators [1,2]. In several plant species, GAs have been directly associated with modulation responses to biotic (e.g., predation and infection) and abiotic stress (e.g., cold, salt, and osmotic stress) [1].

To date, over 136 GA types are described in the literature [3], among which GA3 is the most studied form. Although there are plenty studies on the effects of GA in the field of agriculture and ecology, few articles have analyzed their biological effects in mammals. In a previous study, GA3 was reported to induce the inflammatory regulator A20 in mammal airway epithelial cells, leading to reduced IL-6 and IL-8 release due to NF-κB inhibition [4]. The same scarcity of articles is perceived in relation to microbiology of medical and dental importance. One of the few articles that evaluates this aspect showed that GA4 inhibits a wide spectrum of human clinical bacterial isolates (*Escherichia coli*, *Raoultella planticola*, *Pseudomonas aeruginosa*, *Salmonella nottinglham*, *Bacillus cereus*, *Listeria monocytogenes*, and *Staphylococcus aureus*), while GA7 seems to be less effective, inhibiting the growth of *P. aeruginosa*, *S. nottinglham*, *B. cereus*, *L. monocytogenes*, and *S. aureus* [5]. However, there are no reports in the literature on the anti-*Candida,* anti-NF-κB, or antioxidant effects of GA4 and GA7. The same applies to the toxicity of these compounds.

Thus, in this study, we investigated the in vitro biological effects of commercial GA4 and GA7 compounds, focusing on their anti-NF-κB, anti-*Candida*, and antioxidant activity, and further determined their toxicity in an invertebrate animal model.

## 2. Materials and Methods

### 2.1. Reagents

GA4 (Figure 1A) and GA7 (Figure 1B) were purchased from Santa Cruz biotechnology (EUA). Amphotericin B (AmB), RPMI 1640, Triton-X, *Escherichia coli* B4 lipopolysaccharide (LPS), Fetal Bovine Serum, dextran, penicillin, MTT (3-(4,5-dimethylthiazol-2-yl)-2,5-diphenyltetrazolium bromide), and glutamine were purchased from Sigma-Aldrich (Burlington, MA, USA). 

### 2.2. Effects on NF-κB Activation

#### 2.2.1. Culture Conditions and In Vitro Cell Viability Assay (MTT)

To test the in vitro anti-NF-κB effects of GA4 and GA7, RAW 264.7 macrophages stably transfected with NF-κB luciferase reporter gene (CQB # 022/97) were used. Cells were cultured in RPMI supplemented with 10% FBS, 100 U/mL penicillin, and 2 mM L-glutamine, and incubated at 37 °C in 5% CO_2_ atmosphere. The effects of GA4 and GA7 on macrophage viability were determined by the MTT method. Briefly, cells were seeded into the cavities of 96-well plates (2 × 10^5^ cells/well) and incubated overnight in supplemented RPMI. Cultures were treated with GA4 and GA7 at 30 µM, and incubated in RPMI at 37 °C for 24 h in 5% CO_2_. Later, the supernatant was discarded and 0.3 mg/mL MTT diluted in RPMI was added to the wells and incubated for 3 h. The supernatant was discarded and 200 µL of DMSO was added to each well. The absorbance of the lysates was measured at 570 nm in a microplate reader. The assays were carried out in a quadruplicate of three independent experiments. The mean absorbance of the negative control group was set as 100% viability, whereas GA-treated groups were transformed into percentages of viability according to the formula:% = (A_Treat.GA_ × 100) : M_c-_(1)
where “%” represents the percent viability of the GA-treated well; “A_Treat.GA_” represents the absorbance obtained in the GA-treated well; and “M_c-_” represents the mean absorbance of the negative control. The results were expressed as the mean percent macrophage viability. This viability assay indicates that reductions in luminescence emission (Section 2.2.2) in treated wells are attributed to inhibition of NF-κB activation, and not due to cell death.

#### 2.2.2. In Vitro NF-κB Activation

Macrophages (3 × 10^5^ cells/well) stably transfected with NF-κB luciferase reporter gene were grown in 24-well plates in supplemented RPMI overnight. Then, cells were incubated in the presence of GA4 and GA7 at 3, 10, and 30 µM in RPMI for 30 min at 37 °C in 5% CO_2_. Subsequently, the cultures were stimulated with LPS (10 ng/mL final concentration) for 4 h. The supernatant was collected and discarded, and the remaining cells were used to measure NF-κB activation. Cells were lysed with Tris-NaCl-Tween buffer, and 10 µL of the lysate was mixed with 25 µL of the luciferase assay reagent containing luciferin, and transferred into a 96-well opaque plate. Luminescence was quantified in a microplate reader (SpectraMax^®^ M3, Molecular Devices LLC, Sunnyvale, CA, USA). The results were expressed as relative luminescent units (NF-κB activation level), in relation to the stimulated untreated control group.

#### 2.2.3. Molecular Docking Assays

To investigate the binding capacity of GA4 to NF-κB subunits, we carried out a molecular docking analysis using the AutoDock Vina program [6]. The Uniprot database (UniProt Consortium, 2021) (https://www.uniprot.org/uniprot/ (accessed on: 10 October 2022) was searched for p65 and p50 subunits using the following terms and combinations: gene:rela AND organism: “Mus musculus (Mouse) [10090]” and gene: nfkb1 AND organism: “Mus musculus (Mouse) [10090]”. A new search was performed in the Protein Data Bank [7] to obtain crystallographed or three-dimensionally defined proteins. All proteins were prepared via MGLTools [8], in which all the missing hydrogens were added to the polar atoms, and unnecessary chains and water molecules were removed. The file was converted into the pdbqt format, and the grid box around the docking region was determined. The compound GA4 was retrieved through the PubChem database, and then processed with MGLTools following the same principles for the proteins. Docking was performed in the AutoDock Vina program [6], and the data were visualized using Pymol and LigPlot+ tools.

### 2.3. Anti-Candida Activity

#### 2.3.1. Minimum Inhibitory Concentration (MIC) and Minimum Fungicidal Concentration (MFC) Assays

The anti-*Candida* activity of GA4 and GA7 was determined against *Candida albicans* MYA 2876. The minimum inhibitory concentration (MIC) and minimum fungicidal concentration (MFC) of the compounds were determined by the microdilution method according to Clinical and Laboratory Standards Institute (CLSI) protocols. Briefly, an inoculum of 10^3^ colony-forming units (CFU) per mL was prepared in RPMI medium supplemented with L-glutamine (2 mM), glucose (2%), and MOPS buffer (final pH 7.0). GA4 and GA7 were diluted in RPMI medium and tested at concentrations ranging from 2.9 mM to 188 mM. Amphotericin B (AmB) was used as a positive control at concentrations ranging from 0.54 mM to 1.1 mM. The plates with the inocula and treatments were incubated statically for 24 h. The MIC was determined visually as the lowest concentration of the drug that did not produce any visible turbidity in the wells. Aliquots from the wells with no visible cell growth were plated onto agar sabouraud medium (SDA) plates and incubated for 24 h in an aerobic atmosphere. The MFC was defined as the lowest concentration of the subculture at which no visible growth is observed on solid media. The assays were carried out in triplicate via three independent experiments [9].

#### 2.3.2. Effects on Preformed Biofilm

*Candida albicans* MYA 2876 biofilms were grown in 96-well plates by adding 200 μL of inoculum (10^8^ CFU/mL) to YNB media. The plates were incubated statically at 37 °C for 24 h in an aerobic atmosphere for biofilm formation. The supernatant was discarded, and the wells were washed to remove non-adhered cells. GA7 or AmB (positive control) were prepared in YNB at 2×MIC or 10×MIC (AmB-2×MIC = 1.1 mM and 10×MIC = 5.4 mM; GA7-2×MIC = 188 mM and 10×MIC = 940 mM). GA4 was not tested against the *C. albicans* biofilm as it showed no antimicrobial activity. Biofilms were treated at 37 °C for 24 h in an aerobic atmosphere. The supernatants were discarded, and the wells were washed to remove non-adhered or killed cells. Biofilms were homogenized, serially diluted, and plated onto SDA for CFU counting. The assays were carried out in triplicate via three independent experiments [9]. The results were expressed as percent survival in relation to the untreated group.

### 2.4. Antioxidant Activity

#### 2.4.1. Peroxyl Radical Scavenging Assay (ORAC-ROO•)

The peroxyl radical (ROO•) scavenging activity of the gibberellins was determined according to previous authors [10]. Briefly, aliquots of 30 μL of GA4 and GA7 (23 to 727 mM), 60 μL of fluorescein disodium salt (508 nM), and 110 μL of AAPH (76 mM) were transferred to a microplate to a final volume of 200 μL. All solutions were prepared with 75 mM potassium phosphate buffer (pH 7.4), which was used as a blank. The reaction was performed at 37 °C and absorbance was measured every minute for 2 h at 485 nm (excitation) and 528 nm (emission) in a microplate reader (SpectraMax^®^ M3, Molecular Devices LLC, Sunnyvale, CA, USA). Trolox was used as a standard (12.5 μM to 400 μM), and the results are expressed as μmol Trolox equivalents per mg of GA4 or GA7 (μmol TE/mg).

#### 2.4.2. Superoxide Anion (O_2_^•−^)

NADH (166 µM), nitrotetrazolium blue chloride (NBT) (107.5 µM), GA4 or GA7 (23 mM to 727 mM), phenazine methosulfate (PMS) (2.7 µM), and the diluent potassium phosphate buffer (19 µM, pH 7.4) were added to a final volume of 300 µL. The assay was carried out at 25 °C, and absorbance was measured at 560 nm after 5 min. Potassium phosphate buffer was used as a blank. Trolox was used as a standard (12.5 μM to 400 μM), and the results are expressed as IC_50_, which indicates the minimum amount (mg/mL) of GA4 or GA7 required to quench 50% of the superoxide radicals [11].

#### 2.4.3. Hypochlorous Acid (HOCl)

Briefly, GA4 or GA7 solution (23 μM to 727 mM) homogenized in phosphate buffer (100 mM; pH 7.4), DHR (1.25 µM diluted in 1.15 mM dimethylformamide), and HOCl (5 µM diluted in 1% sodium hypochlorite (NaOCl) solution, adjusted to 6.2 with H_2_SO_4_) were added to a final volume of 300 µL into a 96-well microplate. Fluorescence was measured immediately in a microplate reader (SpectraMax^®^ M3, Molecular Devices LLC, Sunnyvale, CA, USA) with emission at 528 ± 20 nm and excitation at 485 ± 20 nm. The results are expressed as IC_50_ (μg/mL) of GA4 or GA7 [11].

#### 2.4.4. Reactive Nitrogen Species (RNS)

Briefly, 50 μL of GA4 or GA7 solution (23 mM to 727 mM) were added to a 96-well microplate with 50 μL of sodium nitroprusside (SNP) solution (10 mM), 50 μL of phosphate buffer (100 mM; pH 7.4), and 50 μL of diaminofluorescein (DAF) solution (25 μM). Fluorescence was measured in a microplate reader (SpectraMax^®^ M3, Molecular Devices LLC, Sunnyvale, CA, USA) with excitation at 495 nm and emission at 515 nm. Measurements were taken over a 120 min period at 5 min intervals. The results are expressed as IC_50_ (μg/mL) of GA4 and GA7 [10].

### 2.5. In Vivo Toxicity in Galleria mellonella Model

To determine the in vivo toxicity of GA4 and GA7, we carried out a survival assay on *Galleria mellonella* (Phylum: Arthropods, Class: Insect, Order: Lepidoptera, Family: Pyralidae, Subfamily: Galleriinae, Tribe: Gallerini). Larvae were grown at 37 °C, and those with no signs of melanization, and weighing 200 to 300 mg, were randomly selected (n = 10/group). Each larva received 10 μL of GA4 or GA7 at 58.8 mM, 117.5 mM, 235.0 mM, 470.0 mM, and 940.0 mM (0.63 g/Kg, 1.25 g/Kg, 2.5 g/Kg, 5 g/Kg, and 10 g/Kg doses, respectively) or saline solution (control group) in the hemocoel via the last left proleg using a 10 μL Hamilton syringe. All groups were incubated at 37 °C, and survival was monitored at selected intervals for up to 72 h. Larvae showing high melanization and absence of movements upon touch were counted as dead [12].

### 2.6. Statistical Analysis

Data were checked for homoscedasticity and normality, and showed normal distribution. NF-κB activation and biofilm data were analyzed by one-way analysis of variance (ANOVA) followed by Tukey’s post hoc test, and cell viability data were analyzed by unpaired Student’s t-test. The percent survival of *G. mellonella* larvae was compared using the log-rank Mantel−Cox test. In all assays, the sample size was calculated a priori considering an 80% statistical power and a 5% α error. The data were analyzed in GraphPad Prism, version 6.0 for Windows (GraphPad Software, Inc., San Diego, CA, USA).

## 3. Results and Discussion

This study reports on the anti-NF-κB, anti-*Candida*, and antioxidant properties of commercially available gibberellin A4 (GA4) and gibberellin A7 (GA7). Briefly, our data show that GA4 significantly reduced NF-κB activation, whereas GA7 exhibited anti-*Candida* activity against *Candida*
*albicans*, with low MIC/MBC values and antibiofilm properties. Moreover, GA4 and GA7 showed peroxyl radical scavenging activity, and did not affect the survival of *Galleria mellonella* larvae, even at extremely high doses (10 g/Kg).

Figure 2 shows the anti-NF-κB effects of GA4 and GA7 on RAW 264.7 macrophages. Cells were treated with GA4 and GA7 at concentrations below 30 µM, since this was the highest nontoxic concentration measured by the MTT method (Figure 2B,D). After nontoxic treatments, cells were stimulated with LPS at 10 ng/mL, and NF-κB activation was determined by the luminescence assay. LPS stimulus increased NF-κB activation 10-fold. Cells treated with GA4 at 30 µM showed a 32% reduction in NF-κB activation after LPS stimulation (Figure 2A). In contrast, treatment with GA7 did not effectively inhibit NF-κB activation at the tested concentrations in LPS-stimulated cells (Figure 2C).

A molecular docking assay was further carried out to confirm the binding capacity of GA4 to the p65 and p50 subunits of NF-κB. The five best results were selected, and are shown in Table 1.

GA4 interacted with the protein 2LWW_model_1 (Figure 3A) via hydrogen bonds with amino acids Met483 (one hydrogen bond, 3.09 Å) and Ser484 (three hydrogen bonds, 78 Å, 2.90 Å, and 3.17 Å). Moreover, five other amino acids of the protein showed hydrophobic interactions with GA4. As shown in Figure 3B, GA4 also interacted with the 2LWW_model_18 protein via two hydrogen bonds with amino acid THR463 (3.04 Å and 2.82 Å) and via hydrophobic interactions with five amino acids. Similar docking data were observed for 1MY7 (Figure 3C) and 1MY5 proteins (Figure 3D). GA4 interacted with the same amino acid (Arg236) in both proteins, with only a small positional difference, via the hydrogen bonds of 3.01 Å in 1MY5 and 3.08 Å in 1MY7. Lastly, GA4 interacted with the representative p50 protein (1U36) via two hydrogen bonds (2.86 Å and 3.13 Å) with amino acid Phe298 and one hydrogen bond (2.95 Å) with amino acid Gly29. Moreover, the hormone showed binding affinity to four other amino acids of 1U36 via hydrophobic interactions (Figure 3E).

NF-κB is a mediator of an intracellular signaling pathway that is vital to human life, especially for inflammation and immunity responses [13]. Yet, the NF-κB pathway is also related to the onset and development of chronic diseases, such as cancer [14], periodontal disease [15], and rheumatoid arthritis [16]. Therefore, controlling exacerbated NF-κB activation seems to be an effective approach to managing inflammatory diseases [17].

Several external and internal stimuli can trigger the activation of the NF-κB pathway, such as lipopolysaccharides (LPS) from Gram-negative bacteria, and TNF-α and IL1β, respectively [18]. Once activated, NF-κB migrates to the cell nucleus and alters gene expression to prepare for the killing of infectious agents [17]. Macrophages are key cells in these processes since they signal and recruit other cells to the inflammatory focus, such as neutrophils in the acute inflammatory phase [19]. However, on the occasion of chronic inflammatory diseases, the NF-κB inflammatory response is escalated and promotes more damage than benefits.

Periodontal disease is a condition characterized by an exacerbated inflammatory response, which is highly variable in each individual, as a result of an oral microbial dysbiosis [20]. This condition affects the supporting structures of the teeth, and may lead to tooth loss. Consistent with this, several authors have shown that periodontal inflammation is closely related to NF-κB-activation levels. Inflammatory mediators associated with NF-κB activation (such as interleukin-1, TNF-α, and RANKL) are upfolded in diseased periodontal tissues [15]. Therefore, the search for new compounds with anti-NF-κB activity is of great interest for dental practice. Our findings indicate that GA4 has the potential to control the inflammatory process in periodontal disease. In addition, the present study encourages in vivo and clinical trials to confirm the potential of this compound to treat or prevent periodontal diseases.

Table 2 shows the MIC and MFC values of GA4, GA7, and AmB against *C. albicans*. GA4 did not show any inhibitory (MIC) or killing (MFC) effects at the tested concentrations, whereas GA7 showed MIC and MFC values of 94 mM and 188 mM, respectively. As expected, the gold standard AmB showed low MIC and MFC values (0.54 mM and 1.16 mM, respectively), which were lower than those of GA7.

As shown in Figure 4, GA7 effectively reduced the number of viable biofilm cells by 30% at 10×MIC (940 mM). Interestingly, treatment with GA7 at 10×MIC was as effective as AmB (gold standard) at 2×MIC (29.7% reduction in viable biofilm cells). GA7 was not effective at 2×MIC (188 mM), and AmB at 10×MIC reduced CFU counts by 68%.

*Candida albicans* infections play an important role in the development of systemic infectious diseases [21]. *C. albicans* cells colonize most healthy individuals and are often recovered from the dorsum of the tongue. As long as the individual’s immunity is functional, the presence of yeasts tends not to cause great harm; however, immunocompromised patients may suffer from candidiasis. *Candida albicans* oral infection causes discomfort, pain, loss of taste, difficulty swallowing food and liquid, and poor nutrition. However, depending on the severity of the immunosuppression, systemic infections may develop and progress into severe and fatal conditions in 79% of cases due to lack of effective treatment [21,22] In these cases, yeasts form complex biofilms, which are a form of cellular organization between microorganisms that protect resident cells from the host’s immune response and antimicrobial substances [22,23]. Therefore, the ability of GA7 to reduce the burden of viable cells within a biofilm structure is highly relevant.

Our study hypothesis was conceived after we detected the occurrence of GA7 in a Brazilian propolis sample that showed anti-*Candida* activity [24]. This peculiar type of propolis was produced under organic conditions, that is, in a uncontrolled environment where plants may increase the production of GA to resist biotic and abiotic stress, among other possibilities. Therefore, the anti-*Candida* activity of GA7 observed in our study led us to reason that gibberellins are responsible, at least in part, for the anti-*Candida* properties of their originating Brazilian organic propolis extract. However, other molecules may act synergistically with GA regarding the antibiofilm properties of the extract.

The antioxidant assay showed that both compounds were effective against the ROO● radical, named peroxyl (220 ± 50 μmol TE/g for GA4 and 610 ± 40 μmol TE/g for GA7), but not against superoxide anion (O_2_•^−^), hypochlorous acid (HOCl), or nitrogen species (RNS). The peroxyl radical is generated in biological systems from lipid peroxidation. An example of this are the polyunsaturated fatty acids present on the surface of most mammal cells that work as key factors for oxidative stress [25]. Peroxyl is also formed from nonlipid substrates, such as proteins [26]. Once formed, peroxyl radicals trigger diverse inflammatory pathways, and are linked to disease development, such as neurodegenerative and cardiovascular disorders [27,28]. Therefore, the combined antioxidant and anti-NF-κB effects of GA4 present in organic foods are very promising for promoting health and preventing inflammatory diseases. Treatment with GA4 and GA7 did not reduce the percent survival of the larvae at any tested dose during the experimental period (72 h), with no significant difference compared to the control group (saline solution). The *G. mellonella* model is a low-cost, ethical, and reliable method that provides preliminary evidence on the toxicological profile of natural compounds before their further experimentation in other animal models, such as mammal models [12]. To adapt to in vivo conditions, the doses injected into the larvae were at least 10 times higher than the anti-NF-κB (GA4) and anti-*Candida* (GA7) concentrations determined in vitro. Of note, the injected doses of GA4 and GA7 were extremely high and impossible to be consumed by humans; even so, the compounds showed negligible toxicity in this model. Therefore, we can infer that GA4 and GA7 consumption is likely to be nontoxic to mammals, which warrants further investigation.

Thus, our findings indicate that GA4 and GA7 have potential for application in the treatment and prevention of important diseases of the oral cavity. GA4 has potential for application in treating periodontal disease, a disease that generates a loss of dental support tissues (such as gums, periodontal ligament, and alveolar bone), and which, in addition to being associated with changes in the oral bacterial flora, has a strong inflammatory characteristic and notorious participation in the NF-κB pathway [15]. Likewise, GA7 has potential for application in the treatment of oral mycoses and systemic *Candida* infections, since this type of infection is commonly associated with *C. albicans*, an opportunistic fungus that causes both oral and skin infections, but may become invasive and promote systemic candidiasis in immunocompromised hosts [20]. In addition, both compounds showed interesting antioxidant activities against peroxyl radicals and low toxic profiles in the *G. mellonella* model.

## 4. Conclusions

To conclude, GA4 showed inhibitory effects on NF-κB activation (in vitro) and binding affinity to NF-κB subunits (molecular docking); GA7 exhibited anti-*Candida* activity against planktonic and biofilm *Candida albicans* cells. Both GA4 and GA7 showed a significant capacity to scavenge the free radical, peroxyl, and were nontoxic to *Galleria mellonella* larvae, even at extremely high doses. Collectively, the data show that GA4 and GA7 have biological activities of great interest to future application in medical and dental therapy.

## Figures and Tables

**Figure 1 pharmaceutics-14-01347-f001:**
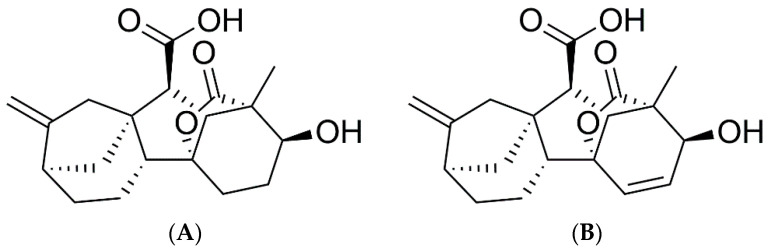
Chemical structures of gibberellin A4 (**A**) and gibberellin A7 (**B**).

**Figure 2 pharmaceutics-14-01347-f002:**
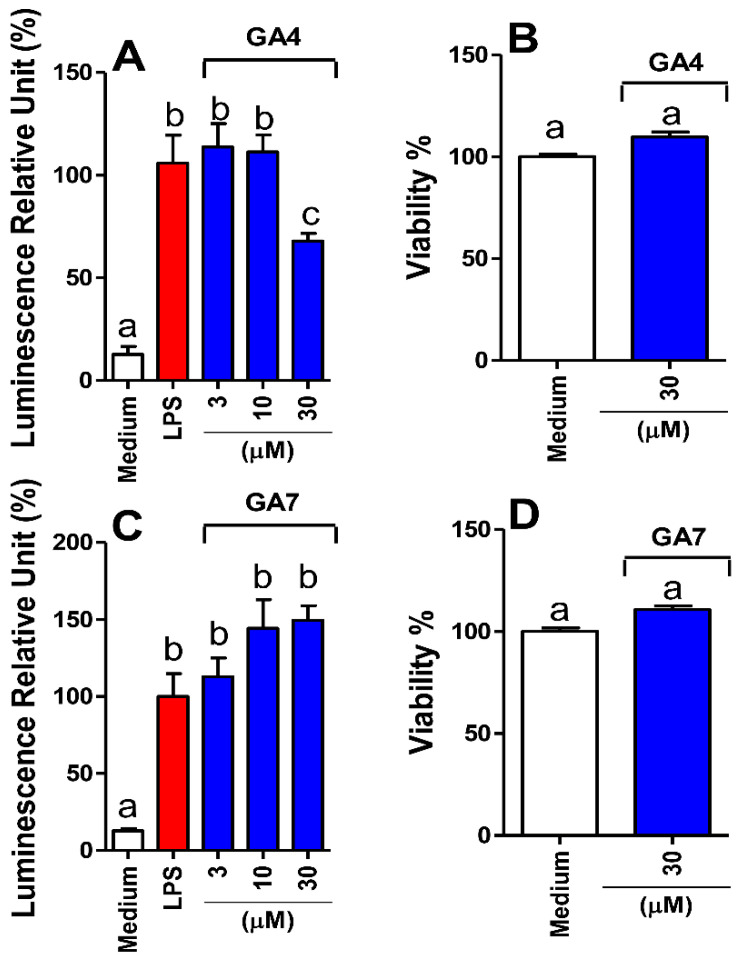
GA4 showed anti-NF-κB activity by reducing NF-κB activation without affecting macrophage viability, while GA7 was ineffective. RAW 264.7 macrophages were treated with GA4 and GA7 at 3, 10, and 30 µM and stimulated with LPS (10 ng/mL). NF-κB activation was measured by the luminescence assay (**A**,**C**). Both GA4 and GA7 did not affect the viability of macrophages by the MTT method (**B**,**D**). Different letters indicate statistically significant differences determined by one-way ANOVA with Tukey’s post hoc test (**A**,**C**) or unpaired *t*-test (**B**,**D**) (*p* < 0.05).

**Figure 3 pharmaceutics-14-01347-f003:**
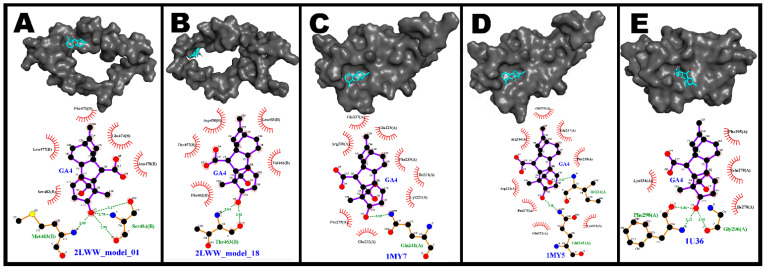
3D and 2D diagrams of the binding between the plant hormone gibberellin A4 (GA4) and the proteins 2LWW_model_01 (**A**), 2LWW_model_18 (**B**), 1MY7 (**C**), 1MY5 (**D**), and 1U36 (**E**).

**Figure 4 pharmaceutics-14-01347-f004:**
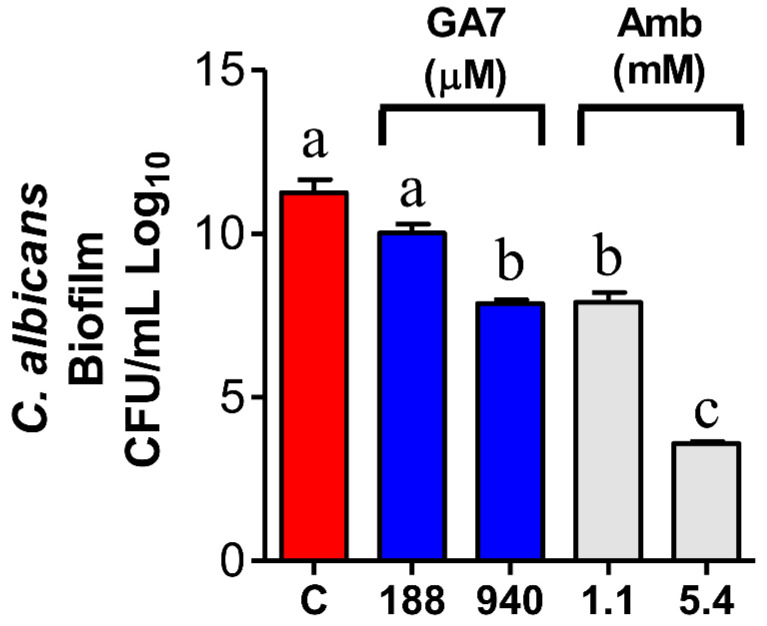
Treatment with GA7 at 10×MIC reduced the viability of *C. albicans* biofilm cells. *C. albicans* biofilms were grown and treated with GA7 or AmB at 2× or 10×MIC. C—Control negative. The results are expressed as CFU/mL. Different letters indicate statistically significant differences (one-way ANOVA with Tukey’s post hoc test, *p* < 0.05).

**Table 1 pharmaceutics-14-01347-t001:** Molecular docking assay with the plant hormone gibberellin A4 (GA4) and the p65 and p50 subunits of NF-κB.

Subunits of NF-κB	Compound	Binding Energy (Kcal·mol^−1^)	Polar Interactions	Non-Polar Interactions	H Bond
p65	2LWW_model_01_GA4	−7.0	2	5	4
p65	2LWW_model_18_GA4	−6.9	1	5	2
p65	1MY7_chain_B_GA4	−6.8	3	5	2
p65	1MY5_chain_B_GA4	−6.5	2	5	1
p50	1U36_GA4	−6.5	1	4	3

**Table 2 pharmaceutics-14-01347-t002:** Minimum inhibitory concentration (MIC) and minimum fungicidal concentration (MFC) values of amphotericin B (Amb), gibberellin A4 (GA4), and gibberellin A7 (GA7) against *C. albicans* MYA 2876.

Treatment Group	MIC (mM)	MFC (mM)
Amb	0.54	1.16
GA4	>188	>188
GA7	94	188

## Data Availability

Not applicable.

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
