# Peer review of "A Study on the Anti-NF-κB, Anti-Candida, and Antioxidant Activities of Two Natural Plant Hormones: Gibberellin A4 and A7"

_pharmaceutics, 2022, doi:10.3390/pharmaceutics14071347_

Round 1

Reviewer 1 Report

Manuscript pharmaceutics-1687356 entitled “A study on the anti-NF-кB, antifungal, and antioxidant activity of gibberellins A4 and A7 - natural plant hormones” provides some preclinical evidence about the bioactivities of GA4 and GA7.  The manuscript is generally well-written. The following items should be either corrected, added, or clarified:

1.       Page 1, line 22: “and antioxidant activity of GA type A4 (GA4) and A7 (GA7) compounds”, please correct to “and antioxidant activity of Gibberellins A4 (GA4) and A7 (GA7) compounds”

2.       Page 2, lines 51-52: “Thus, in this study, we investigated the in vitro biological effects of commercial GA A4 (GA4) and GA A7 (GA7) compounds,” please correct to “Thus, in this study, we investigated the in vitro biological effects of commercial GA4 and GA7 compounds,”

3.       Page 2, section 2.2.1. Culture conditions and in vitro cell viability assay (MTT): please add the equation for calculating survival rate (%).

4.       Page 2, section 2.2.1. Culture conditions and in vitro cell viability assay (MTT): did the author investigate the potential time-dependent effect of GA4 and GA7 on cell viability?

5.       Figure 3: the quality of the figure is too low, please resubmit the figure with high quality.

6.       Page 8, line 262 and Figure 4: please correct the conc. units, make sure all the conc. units are consistent in text and the corresponding figure.

7.       Figure 4: please correct the unit label on the y-axis.

8.       To ensure that the antifungal activity of GA7 is not strain-specific, please provide the antifungal activity of GA7 against other strains of C.albicans? In addition to this, did the author investigate the effect of GA7 on pre-formed biofilm? 

Author Response

Ref. answers to reviewer #1 (pharmaceutics-1687356)

We appreciate the review of our manuscript (pharmaceutics-1687356) and the suggestions given to make it suitable for Pharmaceutics high-quality standards. Please, find below the responses to the reviewers’ comments. We inform you that the manuscript has been revised as per request and all changes are marked-up in red in the text as indicated by the journal.

_______________________________________________________________________________

RESPONSE TO REVIEWER #1:

Manuscript pharmaceutics-1687356 entitled “A study on the anti-NF-кB, antifungal, and antioxidant activity of gibberellins A4 and A7 - natural plant hormones” provides some preclinical evidence about the bioactivities of GA4 and GA7.  The manuscript is generally well-written. The following items should be either corrected, added, or clarified:

  1. Page 1, line 22: “and antioxidant activity of GA type A4 (GA4) and A7 (GA7) compounds”, please correct to “and antioxidant activity of Gibberellins A4 (GA4) and A7 (GA7) compounds”

Answer: We appreciate the correction; the changes have been added to the manuscript text.

  1. Page 2, lines 51-52: “Thus, in this study, we investigated the in vitro biological effects of commercial GA A4 (GA4) and GA A7 (GA7) compounds,” please correct to “Thus, in this study, we investigated the in vitro biological effects of commercial GA4 and GA7 compounds,”

Answer: We appreciate the correction; the changes have been added to the manuscript text.

  1. Page 2, section 2.2.1. Culture conditions and in vitro cell viability assay (MTT): please add the equation for calculating survival rate (%).

Answer: We have changed the text to improve the explanation and have inserted the equation, please check the changes in lines 75 to 82.

  1. Page 2, section 2.2.1. Culture conditions and in vitro cell viability assay (MTT): did the author investigate the potential time-dependent effect of GA4 and GA7 on cell viability?

Answer: In the viability experiment (session 2.2.1), we aimed to verify if macrophages remain viable throughout the treatment with gibberellins, thus ensuring reductions in luminescence emission (session 2.2.2) in treated wells attributed to a reduction in NF-кB activation and not to cell mortality. Thus, the viability study took place with a treatment time longer (24 hours) than the treatment time performed in the NF-кB experiment (4 hours). Thus, our results presented in Figure 2 can be attributed to a reduction in NF-кB activation. To make this reasoning clearer in the manuscript text, we added lines 82 to 84 in session 2.2.1.

  1. Figure 3: the quality of the figure is too low, please resubmit the figure with high quality.

Answer: A higher resolution figure file was attached to the mdpi platform and also to the manuscript itself.

  1. Page 8, line 262 and Figure 4: please correct the conc. units, make sure all the conc. units are consistent in text and the corresponding figure.

Answer: We apologize for the misunderstanding, we altered the paragraph (lines 276 to 279) and now the text fits the figure.

  1. Figure 4: please correct the unit label on the y-axis.

Answer: We apologize for the misunderstanding, we carry out the correction in the figure.

  1. To ensure that the antifungal activity of GA7 is not strain-specific, please provide the antifungal activity of GA7 against other strains of C.albicans? In addition to this, did the author investigate the effect of GA7 on pre-formed biofilm?

Answer: We appreciate the reviewer's comments. This is the first report of the biological activities of GA7, therefore, at this moment we made a preliminary screening of the biological activities of the compound. We chose a commercial strain that is representative of the Candida albicans species (MYA 2876) and in the future, we will be able to look for activities in other strains and also explore mechanisms of action that explain the anti-Candida activities presented in this manuscript. To make this reasoning clearer, we changed the manuscript text, switching the term “antifungal” to the term “anti-Candida”. Regarding the question about the pre-formed biofilm, the study we carried out evaluated the effect of GA7 on the C. albicans mature biofilm. We believe that the title of subsection 2.3.2. was ambiguous generating doubts in the readers, who could incorrectly understand that the assay was done with biofilm in formation. To correct the ambiguity, we have changed the title of this subsection to “Effects on preformed biofilm”.

In addition to the previous answers, the use of the English language in the manuscript has been fully reviewed by a professional, as described in the attached voucher. Therefore, we expect to have made all the corrections in the manuscript according to the reviewers’ comments, which contributed enormously to improving the quality of our manuscript. We believe that, as it stands, the manuscript satisfactorily meets the high standards required for publication in the Pharmaceutics.

Sincerely yours,

Prof. Dr. Marcelo Franchin

School of Dentistry

Federal University of Alfenas (Unifal-MG)

Reviewer 2 Report

Nicely written study

Author Response

Ref. answers to reviewer #2 (pharmaceutics-1687356) 

We appreciate the review of our manuscript (pharmaceutics-1687356) and the suggestions given to make it suitable for Pharmaceutics high-quality standards. Please, find below the responses to the reviewers’ comments. We inform you that the manuscript has been revised as per request and all changes are marked-up in red in the text as indicated by the journal.

_________________________________________________________________________

RESPONSE TO REVIEWER #2: “Nicely written study.”

We thank the reviewer for the acceptance. Furthermore, the use of the English language in the manuscript has been fully reviewed by a professional, as described in the attached voucher.

Therefore, we expect to have made all the corrections in the manuscript according to the reviewers’ comments, which contributed enormously to improving the quality of our manuscript. We believe that, as it stands, the manuscript satisfactorily meets the high standards required for publication in the Pharmaceutics.

Yours sincerely,

Prof. Dr. Marcelo Franchin

School of Dentistry

Federal University of Alfenas (Unifal-MG)

Reviewer 3 Report

The text is well written and the experimental part well described. However, it is no clearly explained how the interesting in vitro anti-NF-кB, antifungal and antioxidant activities of GA type A4 and A7 are connected to the possible use for the treatment and prevention of diseases of the oral cavity, such as periodontitis. This point should be more developed in the paragraph between lines 316-325 as well as in the Introduction part and could support the biological interest and possible medical applications of these compounds.

Author Response

Ref. answers to reviewer #3 (pharmaceutics-1687356) 

We appreciate the review of our manuscript (pharmaceutics-1687356) and the suggestions given to make it suitable for Pharmaceutics high-quality standards. Please, find below the responses to the reviewers’ comments. We inform you that the manuscript has been revised as per request and all changes are marked-up in red in the text as indicated by the journal.

________________________________________________________________________

RESPONSE TO REVIEWER #3:

“The text is well written and the experimental part well described. However, it is no clearly explained how the interesting in vitro anti-NF-кB, antifungal and antioxidant activities of GA type A4 and A7 are connected to the possible use for the treatment and prevention of diseases of the oral cavity, such as periodontitis. This point should be more developed in the paragraph between lines 316-325 as well as in the Introduction part and could support the biological interest and possible medical applications of these compounds.”

Answer: We agree that it is necessary to include more information on periodontal disease and candidiasis in the text to could support the biological interest and possible medical applications of compounds. Therefore, we have included the paragraphs present in lines 256– 266 and 284 – 295.

Furthermore, the use of the English language in the manuscript has been fully reviewed by a professional, as described in the attached voucher. Therefore, we expect to have made all the corrections in the manuscript according to the reviewers’ comments, which contributed enormously to improving the quality of our manuscript. We believe that, as it stands, the manuscript satisfactorily meets the high standards required for publication in the Pharmaceutics.

Yours sincerely,

Prof. Dr. Marcelo Franchin

School of Dentistry

Federal University of Alfenas (Unifal-MG)

Reviewer 4 Report

The authors created a study that examined several properties of the gibberellins A4 and A7, and they concluded a potential pharmaceutical use based on their results.  The text reads well but contains "specialist's jargon" that should be avoided. Comments and proposals for changes can be found in the document uploaded by the reviewer.

Author Response

Ref. answers to reviewer #4 (pharmaceutics-1687356) 

We appreciate the review of our manuscript (pharmaceutics-1687356) and the suggestions given to make it suitable for Pharmaceutics high-quality standards. Please, find below the responses to the reviewers’ comments. We inform you that the manuscript has been revised as per request and all changes are marked-up in red in the text as indicated by the journal.

_________________________________________________________________________

RESPONSE TO REVIEWER #4:

“The authors created a study that examined several properties of the gibberellins A4 and A7, and they concluded a potential pharmaceutical use based on their results.  The text reads well but contains "specialist's jargon" that should be avoided. Comments and proposals for changes can be found in the document uploaded by the reviewer.”

Answer: We appreciate the outstanding comments and suggestions which were all accepted. The changes have been added to the manuscript. Furthermore, the use of the English language in the manuscript has been fully reviewed by a professional, as described in the attached voucher. Therefore, we expect to have made all the corrections in the manuscript according to the reviewers’ comments, which contributed enormously to improving the quality of our manuscript. We believe that, as it stands, the manuscript satisfactorily meets the high standards required for publication in the Pharmaceutics.

Yours sincerely,

Prof. Dr. Marcelo Franchin

School of Dentistry

Federal University of Alfenas (Unifal-MG)
